# Reconstructing Rural Settlements Based on Structural Equation Modeling—Taking Hongshanyao Town of Jinchang City as an Example

**Xiaoling Xie * and Lin Ye**

School of Architecture and Urban Planning, Lanzhou Jiaotong University, Lanzhou 730070, China
* Correspondence: xxl00117@163.com

**Abstract:** Rapid urbanization has driven great changes in China's rural areas. In order to adapt to the changes in the internal elements and external regulation of the countryside, rural reconfiguration, i.e., to make rural development more adaptable to the spatial pattern of social development through optimal allocation and effective management, can achieve structural optimization and functional improvement within the rural territorial system. This study selects Hongshanyao Town of Jinchang City as the study area, constructs a structural equation model and an adaptability evaluation system to investigate the suitability of rural settlement layout, and constructs a differentiated and suitable rural settlement reconstruction model for different villages based on meeting farmers' wishes, to intending to serve the current major national strategic needs and solve the real dilemmas faced by rural areas in the process of urban-rural transformation and development. The corresponding strategies are proposed, which provide a theoretical basis for future village development and spatial reconfiguration practices in rural areas, and are of great significance for realizing rural revitalization as well as village planning and construction.

**Keywords:** structural equation model; suitability evaluation; rural settlement; reconstruction model; Hongshanyao town

## 1. Introduction

Rural reconfiguration is associated with the decline of the countryside that accompanies globalization and urbanization [1]. Since the 1950s, the social structure and ecological environment of rural areas in developed countries such as the United Kingdom and the United States have undergone significant changes after urbanization and counter-urbanization, and since the 1980s, the socio-economic development of China has led to a gradual change in the traditional characteristics of the countryside with the process of urbanization and industrialization. In recent years, rural reconfiguration has become a hot issue in Chinese academia, and experts and scholars from various fields, including geography and sociology, have reflected the results of contemporary Chinese rural reconfiguration research based on different perspectives and using different research methods, research ideas, and reconfiguration types to focus on the spatial form, regional functions, and the reshaping and optimization of the governance system of the countryside [2]. In terms of research methods, the early studies mainly focused on the morphology of the physical space of villages [3], and then emphasized the non-physical environment of villages, and the research on the process tended to adopt dynamic quantitative methods, by adopting a combination of quantitative evaluation indicators and development potential evaluation models, constructing a village potential evaluation system, determining the weights of influencing factors and a quantitative village classification method with a residential density as a single criterion, and using GIS spatial analysis method to extract the settlement density of rural settlements, screen the villages of different classes, and then verify the results of village potential evaluation utilizing field surveys and other means, to classify the

villages [4,5]. The research idea is to explore the combination of characteristics of livability and population flow for different development modes of villages, and to reflect the idea of village classification by retracing the construction history of villages through the planning framework of "spatial classification + planning mode". According to the logic of "resource efficiency", the potential of village revitalization and land use efficiency is taken as the basis of judgment; in terms of reconstructing types, villages are divided into different types by analyzing the characteristics of population flow, residential density, land use efficiency and village development potential, among which, villages with lower suitability are demolished and annexed, while villages with high population flow are more livable [6]. The villages with low suitability are demolition and annexation villages, those with strong livability are gathering and upgrading villages, those with good location conditions are suburban integration villages, and those with rich history and culture are characteristic preservation villages, etc. Thus, it can be seen that under the influence of differentiated development, the classification criteria and bases of villages in different regions are different. Although the results of village classification are supported by technical methods, they lack attention to the development elements of villages themselves to a certain extent [7,8].

In this paper, the method of "structural equation modeling + suitability evaluation" is integrated with farmers' actual feelings and real evaluation of their living places at the township scale, and the construction mode of villages is reconstructed in a differentiated way, which can effectively integrate rural land resources, improve the intensification of land use, improve the structure of villages, optimize the layout of settlements, rationalize the allocation of resources, and improve the development of villages. The layout of settlements and rational allocation of resources can realize rural revitalization.

## 2. Materials and Methods

### 2.1. Research Area

Hongshanyao town is located in the western part of Jinchang City, 40 km from the county, with a total area of 1540 square kilometers. It is a semi-agricultural and semi-pastoral agricultural town, which was withdrawn in 2017 to change the township. It comprises a total of 12 administrative villages and has a total population of 24,542 people,as shown in Figure 1. Hongshanyao town territory belongs to the Shiyang River basin, the terrain is high in the west and low in the east, and the most elevation is located in the village of Maobla. It has a temperate continental climate, arid climate, and low rainfall.

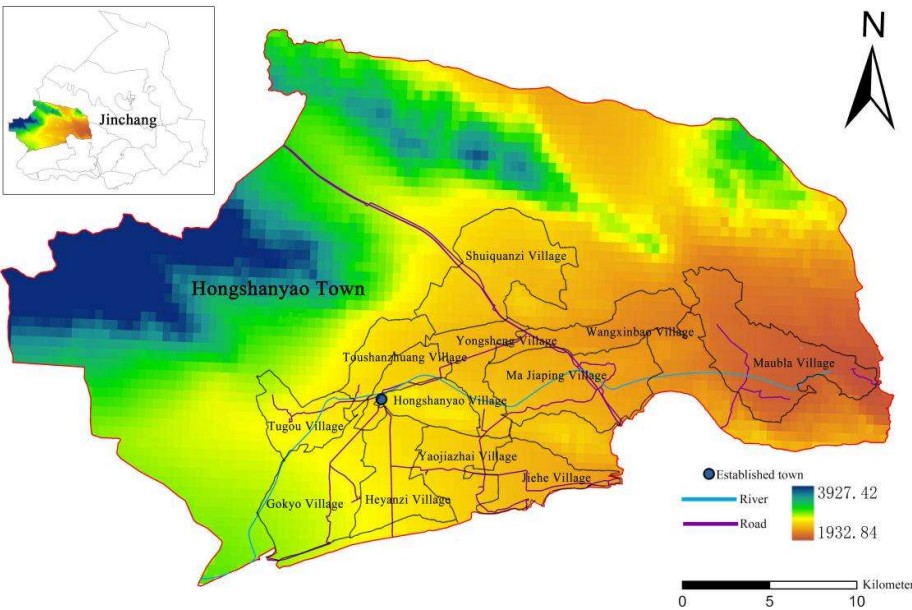

**Figure 1.** Location map of Hongshanyao town.

The reason for choosing Hongshanyao town in this study is that it has the following special features and typicalities: ① Hongshanyao town is located in the western corridor of the river, with deep inland arid area, with complicated topography, high in the west and low in the east, but also the Shiyang River basin, with the Xida River running through its territory, and rural settlements are mostly distributed in the oasis plain, and its location conditions are different from other villages in Jinchang city; ② Hongshanyao town is both a traditional agricultural town and a large livestock town, and it has a high potential for development because of the development of mining resources and the national highway, and it has various historical and cultural resources. To sum up, Hongshanyao town has diversified development bases, advantages, and potentials, and the reconstructed model can be differentiated according to the different characteristics of villages to explore the development model with different characteristics.

### 2.2. Data Description

In this survey, a total of 12 administrative villages in one town were selected as sample points. 600 questionnaires were distributed, and 576 were returned, of which 529 were valid, with a sampling efficiency of 88%, as shown in Table 1 below. In the survey process, to ensure the objectivity and authenticity of the research results, the respondents are all long-term residents of the local residents and ensure that the residence time is more than one year while taking into account the respondents' education level and age differences, the rural residents of different levels were selected to interview.

**Table 1.** Distribution of the survey sample.

| Village | Number of Samples | Effective Sample Size |
|---|---|---|
| Hongshanyao Village | 50 | 48 |
| Shantouzhuang Village | 50 | 45 |
| Shuiquanzi Village | 50 | 42 |
| Yongsheng Village | 50 | 46 |
| Wangxinbao Village | 50 | 41 |
| Ma Jiaping Village | 50 | 42 |
| Jiehe Village | 50 | 48 |
| Yaojiazhai Village | 50 | 43 |
| Heyanzi Village | 50 | 44 |
| Gokyo Village | 50 | 44 |
| Tugou Village | 50 | 45 |
| Maubla Village | 50 | 41 |
| Total | 600 | 0 |

### 2.3. Research Methodology

#### 2.3.1. Model Assumptions and SEM Model Construction

Due to the unique geographical location of Jinchang City, the assessment of the location of rural settlements in Hongshanyao Town is subject to a variety of uncertainties, making it difficult to determine the most influential and important factors [9]. To enhance the stability of the research results, structural equation modeling is used to analyze the measurement error with the help of a set of latent variables that are mainly observed but not directly measured, to verify the impact of the rural habitat environment on rural development. SEM model is a multivariate statistical model, which mainly studies the interaction relationship between variables, and it integrates two methods, factor analysis, and path analysis, to realize the organic combination of measurement and analysis, which can not only It can explore the real relationship between independent variables based on the covariance matrix between variables, and also can deal with multiple variables simultaneously for multi-factor analysis [10,11].

First of all, before the SEM model analysis, the latent variables and observed variables in the model should be clarified. In this paper, three latent variables are determined according to the actual situation, development needs, and development direction of Hongshanyao

town; secondly, the data collected and summarized are substituted into the structural equation model, and the parameters of the model are estimated to verify the degree of conformity between the model and the actual situation and to judge the degree of fitness of the indicators. Finally, the model is modified according to the actual situation [12,13]. The SEM model consists of two parts: the measurement equation and the structural equation, with three covariance matrix equations [14,15]. The study analyzes the relationship between natural, social, economic, and locational factors and the distribution of rural settlements, and their structural equations.

$$\eta = B\eta + \Gamma\xi + \zeta \tag{1}$$

$$X = \Lambda x\xi + \delta \tag{2}$$

$$Y = \Lambda y\eta + \varepsilon \tag{3}$$

$\eta$ is the potential dependent variable, which refers to the distribution location of rural settlements; $\xi$ is the exogenous potential force, which refers to natural, social, and economic factors; $B$ is the relationship between the potential dependent variables; $\Gamma$ is the extent to which the potential independent variable influences the potential dependent variable; and $\zeta$ denotes the correction made for the unexplained potential response variable.

### 2.3.2. Variable Selection

Set the farmers' satisfaction with the development of rural settlements as the explanatory variable, the main body of rural settlements is the farmers, so the survey of satisfaction with residential addresses has authenticity, such as the existing area of housing, housing structure, traffic accessibility and satisfaction with the surrounding services, etc. All these factors will affect the farmers' willingness to relocate (consolidate or relocate), some of which are pulling factors and some of which are resistance [16–18].

The questionnaire of this study combines the actual situation of current rural life and production and follows the principles of scientific rationality, and designs three latent variables such as family characteristics, residence characteristics, and social characteristics, and several observed variables such as education level and building structure [19]. As shown in Table 2 below, the study is based on the five-point Likert scale. At the same time, a five-point Likert scale scoring method was used as the measure of the questionnaire [20–22].

### 2.3.3. Reliability and Validity Tests

Reliability test: As shown in Table 3 below, reliability can reflect the reliability, consistency, and stability of the questionnaire to a certain extent, and is generally measured using the Cronbach alpha coefficient value in SPSS software [23,24]. When the Cronbach alpha coefficient is greater than 0.7, it means that the reliability is high. The Cronbach's alpha coefficient of 0.758 meets the reliability requirement, which indicates that the questionnaire design has high reliability and can be studied in the next step.

Validity test: This study was tested for validity by the KMO (Kaiser Meyer Olkin) test and Bartlett's value with the help of SPSS software. The closer the KMO value is to 1, the more the measurement results match the content of the examination [25]. The Table 4 shows that the value of KMO is 0.845 and the value of Bartlett is 4036.25, which is significantly less than 0.01, indicating that it is suitable for factor analysis and meets the validity test, and further empirical studies can be conducted.

**Table 2.** Factors of satisfaction with farm dwellings.

| Latent Variable | Explicit Variables | Variable Definition and Description |
|---|---|---|
| Family characteristics (F1) | Education level (A1) | 1 = elementary school; 2 = middle school; 3 = high school. 4 = Secondary, college; 5 = Bachelor and above |
| | Occupation (A2) | 1 = farmers; 2 = village officials; 3 = individuals. 4 = public official; 5 = other |
| | Disposable income (A3) | 1 = in short supply; 2 = subsistence; 3 = just right. 4 = Adequate; 5 = Rich |
| | Financial burden (A4) | 1 = very heavy burden; 2 = heavy burden; 3 = average burden; 4 = light burden; 5 = no burden |
| Residence characteristics (F2) | House construction time (B1) | 1 = before 1981; 2 = between 198 and 990; 3 = between 199 and 000; 4 = between 200 and 010; 5 = between 201 and 020 |
| | Building Structure (B2) | 1 = adobe; 2 = adobe structure; 3 = brick structure; 4 = brick and concrete; 5 = steel and concrete |
| | Homestead area per capita (B3) | 1 = very low; 2 = low; 3 = just right. 4 = high; 5 = very high |
| | Residential base utilization (B4) | 0 = no idle; 1 = with idle |
| Social Characteristics (F3) | Traffic accessibility (C1) | 1 = very poorly ventilated; 2 = poorly ventilated; 3 = generally ventilated; 4 = ventilated; 5 = very ventilated |
| | Service facility utilization (C2) | 1 = very dissatisfied; 2 = not satisfied; 3 = fairly satisfied; 4 = satisfied; 5 = very satisfied |
| | Neighborhood (C3) | 1 = very bad; 2 = bad; 3 = fair. 4 = good; 5 = very good |
| | Habitat suitability (C4) | 1 = very unsuitable; 2 = unsuitable; 3 = generally suitable; 4 = suitable; 5 = very suitable |
| | Level of knowledge of policies (C5) | 0 = no knowledge; 1 = knowledge |

**Table 3.** Statistics of main index items.

| Indicators | Average Value | Standard Deviation | Number of Cases |
|---|---|---|---|
| A1 Literacy | 2.73 | 0.984 | 529 |
| A2 Careers | 2.59 | 1.258 | 529 |
| A3 disposable income | 2.68 | 1.130 | 529 |
| A4 Economic burden | 2.75 | 1.149 | 529 |
| B1 house construction time | 2.44 | 1.015 | 529 |
| B2 building structure | 2.42 | 1.021 | 529 |
| B3 per capita home base area | 2.41 | 1.002 | 529 |
| B4 residential base utilization | 0.50 | 0.500 | 529 |
| C1 Traffic accessibility | 2.70 | 1.148 | 529 |
| C2 service facility utilization | 2.72 | 1.171 | 529 |
| C3 Neighborhood | 2.69 | 1.151 | 529 |
| C4 Habitat Suitability | 2.64 | 1.138 | 529 |
| Level of understanding of C5 policy | 0.54 | 0.499 | 529 |

**Table 4.** KMO values and Bartlett's test.

| Name | | Value |
|---|---|---|
| KMO Sampling suitability quantity | | 0.845 |
| Bartlett's sphericity test | Approximate cardinality | 4036.250 |
| | Degree of freedom | 78 |
| | Significance | 0.000 |

### 2.3.4. Suitability Evaluation Index Selection

The suitability evaluation of rural settlements is a comprehensive assessment of the location of settlement construction [26,27]. The indicators are selected from the aspects of nature and location for the current situation of Hongshanyao town. The natural factors mainly consider the influence of elevation and slope on rural settlements, and the location factors mainly consider the influence of rivers and roads on rural settlements, and different evaluation indicators have different degrees of influence on the layout of rural settlements, and their weights are assigned differently [28].

To reduce the mutual influence of different indicators, the indicators are reduced in equal proportion and standardized so that their values are between 0 and 1, and, finally, the indicators in the comprehensive index system can be The weighted sum of the indicators in the comprehensive index system is finally, made possible [29].

$$\text{Positivity.} \quad Pi = \frac{Ti - Tmin}{Tmax - Tmin} \tag{4}$$

$$\text{Negativity.} \quad Pi = \frac{Tmax - Ti}{Tmax - Tmin} \tag{5}$$

The attributes of the indicators are divided into positive and negative, where positive indicators are those that have a positive impact on the spatial distribution of rural settlements. The higher the value of the indicator, the higher the degree of suitability of rural settlements; negative indicators are the opposite the higher the value, the lower the suitability of rural settlements [30].

## 3. Results

### 3.1. Model Construction and Data Analysis

The analysis software Amos28 was used to construct a model of factors influencing the distribution of rural settlements in Hongshanyao town and to evaluate the importance of each factor,as shown in Figure 2. The oval represents the potential variable. The rectangle indicates the measurement index. The circle identifies the error e; the arrow identifies the correlation between the elements. The arrows are marked with the weight coefficients [31–33].

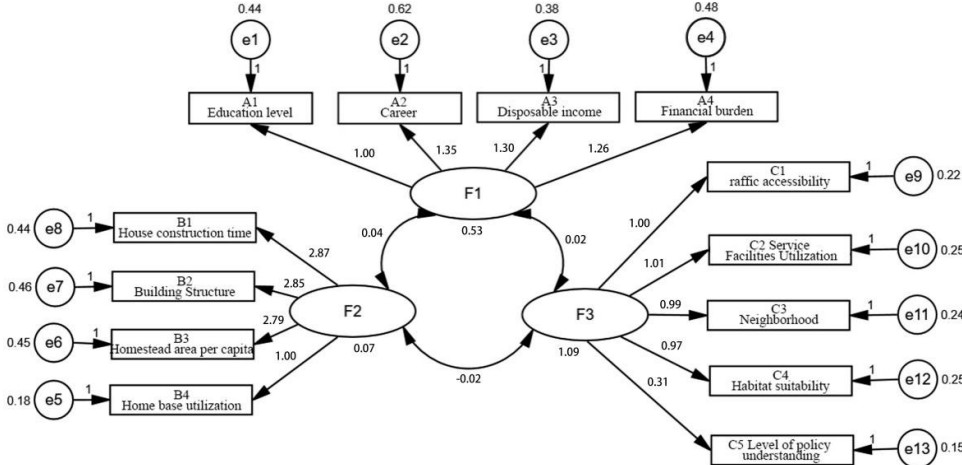

**Figure 2.** Structural equation model. Note: F1 is the household characteristics; Ai is the 4 factors included in the household characteristics; F2 is the residence factors; Bi is the 4 factors included in the residence characteristics; F3 is the social characteristics; and Ci is the 5 factors included in the social characteristics.

By using AMOS software, the initial structural equation model is verified using the research data, and the path is adjusted according to the first fit result, to obtain the structural equation model. As shown in Table 5 below, when the model is evaluated for fit, the higher the fit is, the stronger the model is in explaining the problem, and the evaluation indexes of

the SEM model of Hongshanyao town mainly have three categories, which are absolute fit index, value-added fit index There are three main categories of evaluation indexes for the Hongshanyao town SEM model, namely, absolute fit index, value-added fit index, and simple fit index, in which the goodness-of-fit index is 0.959 over 0.9 and the adjusted fit index is more than 0.9; the comparative fit index is 0.978 over 0.9; and the simple fit adjustment index is 0.777 over 0. By comparing the reference values of evaluation standards, the indicators of the model meet the requirements and the model has a good fit.

**Table 5.** Model Adaptation Metrics.

| Evaluation Category | Reference Standards | Default Model | Saturation Model | Independent Models | Reference Value |
|---|---|---|---|---|---|
| Absolute Adaptation Index | Cardinality CMIN | 150.415 | 0 | 4076.136 | – |
| | Degree of Freedom DF | 62 | 0 | 78 | – |
| | The probability of significance P | 0.072 | | 0 | >0.05 |
| | Goodness-of-fit index GFI | 0.959 | 1 | 0.407 | >0.9 |
| | Adjusted Adaptation Index AGFI | 0.94 | | 0.309 | >0.9 |
| | Difference divided by degrees of freedom CMIN/DF | 1.426 | | 52.258 | <2 |
| | Asymptotic residual mean square and square root RMSEA | 0.032 | | 0.312 | <0.05 |
| Value Added Adaptation Index | Value-added Adaptation Index IFI | 0.978 | 1.0 | 0.0 | >0.9 |
| | Non-standard adaptation index TLI | 0.972 | | 0.0 | >0.9 |
| | Comparative Fit Index CFI | 0.978 | 1.0 | 0.0 | >0.9 |
| Simplicity Adaptation Index | Information school standard AIC | 208.415 | 182 | 4102.136 | – |
| | Minimalist Adjusted Gauge Fit Index PNFI | 0.766 | 0.0 | 0.0 | >0.5 |
| | Minimalist Adaptation Adjustment Index PCFI | 0.777 | 0.0 | 0.0 | >0 |

The path regression coefficients of different factors were calculated by AMOS software to judge the relevance of each factor on the layout of rural settlements in Hongshanyao town. Among the F1 indicators, the standardized regression coefficient of disposable income of farm households is 0.838, which indicates that the income of villagers has a high correlation with the comfort of rural residence, and the higher the income, the stronger the dependence on agricultural production and the more satisfied with the residence; in In F2, the highest standardized regression coefficient of house construction time is 0.756, which shows that the longer the houses are built, the more serious the damage is, the worse the living conditions are, and the stronger the farmers' willingness to relocate and consolidate and renovate their houses. The highest standardized regression coefficient of the F3 indicator is traffic accessibility, which is 0.912, indicating that villagers are more concerned about traffic accessibility and prefer to live in areas with convenient traffic to facilitate travel; looking at the standardized regression coefficients of all indicators, it can be seen that the utilization of house bases (whether they are unused or not) and villagers' understanding of policies are low, indicating that whether the houses are used or not and farmers' policy interpretation have little influence on villagers' evaluation of the suitability of housing, as shown in Table 6 below.

### 3.2. Rural Settlement Suitability Evaluation

This study, after analyzing farmers' satisfaction with rural settlements and judging the factors affecting the layout of rural settlements through structural equation modeling, summarizes the relevant factors affecting the spatial layout of rural settlements according to the current conditions of Hongshan Yao town, and fully considers the comfort, convenience, and other production and living needs of farmers' dwellings, constructs five criterion layers with a total of 18 indicators, and determines each evaluation indicator by comprehensive analysis of different indicators weight of each indicator [34,35]. The weight of each indicator is determined by a comprehensive analysis of different indicators, as shown in Table 7 below.

**Table 6.** Path regression coefficients.

| | Indicators | Unstandardized Estimates | Standard Error S.E. | Critical Ratio C.R. | Standardized Estimates |
|---|---|---|---|---|---|
| F1 | A1 Literacy | 1 | | | 0.738 |
| | A2 Careers | 1.349 | 0.08 | 16.889 | 0.779 |
| | A3 disposable income | 1.303 | 0.073 | 17.898 | 0.838 |
| | A4 Economic burden | 1.265 | 0.072 | 17.560 | 0.800 |
| F2 | B1 house construction time | 2.871 | 0.261 | 11.011 | 0.756 |
| | B2 building structure | 2.851 | 0.263 | 10.850 | 0.747 |
| | B3 per capita home base area | 2.786 | 0.256 | 10.887 | 0.744 |
| | B4 residential base utilization | 1s | | | 0.534 |
| F3 | C1 Traffic accessibility | 1 | | | 0.912 |
| | C2 service facility utilization | 1.01 | 0.03 | 33.319 | 0.904 |
| | C3 Neighborhood | 0.993 | 0.03 | 33.455 | 0.904 |
| | C4 Habitat Suitability | 0.974 | 0.03 | 32.809 | 0.897 |
| 54111′ | C5 Level of understanding of policy | 0.305 | 0.017 | 17.432 | 0.640 |

**Table 7.** Suitability evaluation indicators.

| Target Layer | Guideline Layer | Indicator Layer | Indicator Functions | Weights | Properties |
|---|---|---|---|---|---|
| Comprehensive impact evaluation index system of rural settlements A | Physical Geography B1 | Elevation C1 | Judging the relocation and integration areas of rural settlements based on their innate factors | 0.61 | Negative |
| | | Slope C2 | | 0.31 | Negative |
| | | Slope direction C3 | | 0.23 | Positive |
| | Agricultural development conditions B2 | Arable land per capita C4 | Advantages and disadvantages of agricultural development as the basis for the survival of farmers | 0.42 | Positive |
| | | Food production C5 | | 0.38 | Positive |
| | | Kojubi C6 | | 0.48 | Positive |
| | | Status of agricultural facilities C7 | | 0.36 | Positive |
| | Settlement base condition B3 | The population size of the settlement C8 | Determine the trend of settlement development based on the land and population size of rural residents | 0.23 | Positive |
| | | Settlement site size C9 | | 0.32 | Positive |
| | | Per capita settlement area C10 | | 0.28 | Positive |
| | | Average tillage straight-line distance C11 | | 0.35 | Negative |
| | Location condition B4 | Distance from the main road C12 | Determine the construction location of the settlement with the aid of location conditions | 0.62 | Negative |
| | | Distance from established towns C13 | | 0.42 | Negative |
| | | Distance from village council C14 | | 0.36 | Negative |
| | | Distance to river C15 | | 0.56 | Negative |
| | Comprehensive economic development conditions B5 | Net income per capita C16 | Assist in determining the construction location of the settlement | 0.53 | Positive |
| | | Traffic accessibility C17 | | 0.63 | Positive |
| | | Infrastructure Convenience C18 | | 0.34 | Positive |

This paper synthesizes the above SEM model suitability evaluation analysis and screening and with the help of ArcGIS software, the layout suitability evaluation map of Hongshanyao town is obtained through the overlay analysis in the software, and it is graded into four levels: most suitable, more suitable, generally suitable and unsuitable. The study develops that the suitability of rural settlements is influenced by highways, rivers, and established towns, which show a strip distribution along highways and rivers and disperse around established towns, as shown in Figure 3.

In 2020, there are 66 rural settlements in Hongshanyao Township in the most suitable area, accounting for 64% of the total number of rural settlements and 34.2% of the total area of the township, of which: Tugou Village, Toushanzhuang Village, Hongshanyao Village, Yongsheng Village, Yaojiazhai Village, and Heyanzi Village are located in the most suitable area of the study area, mainly in the flat areas on both sides of the rivers and roads; the more suitable area of the study area has the largest distribution area, accounting for 41.4% of the total area of the township. The most suitable area in the study area is 41.4% of the total area of the town, with 28 settlements, among which the village in the more suitable area is Maobla Village, which is located at the eastern end of the study area and has the lowest terrain and river in the whole study area. There are 8 rural settlements in

the study area, but only Shuiquanzi Village, the northernmost village in the township, has a part of the area in the general suitable zone, and there are roads in the village; there are the least rural settlements in the unsuitable zone, because the topography of the area is complicated, far from roads and rivers, the living conditions are poor and unfavorable to the production and living of farmers. In terms of the distribution of rural settlements in the study area, there are still rural settlements in the western part of the township outside the administrative villages because it is a semi-agricultural and semi-pastoral area where some managers and employees live to facilitate management and work, and spontaneously form settlements.

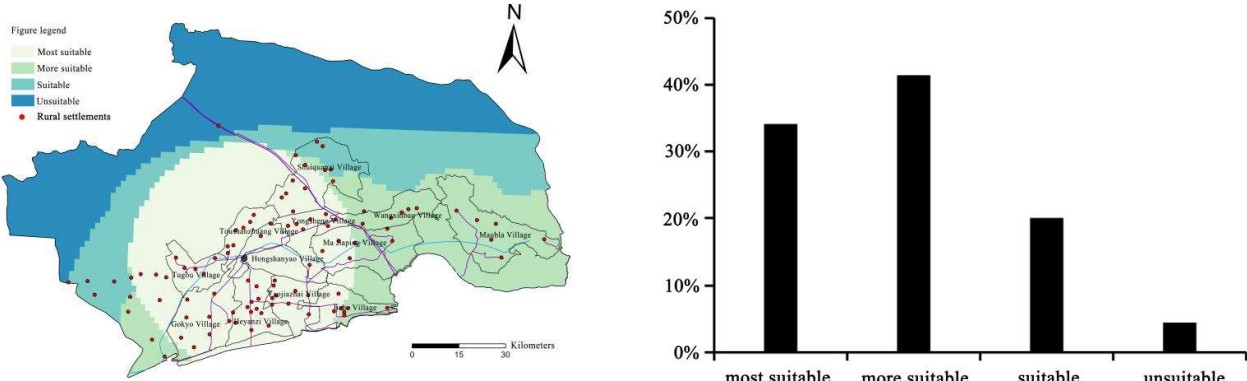

**Figure 3.** Map of settlement suitability and percentage.

## 4. Discussion

### 4.1. Rural Settlement Layout Characteristics

The distribution of rural settlements in Hongshanyao town in 2020 is affected by the topography with the characteristic of "sparse in the north and dense in the south", and the distribution of rural settlements in the south of the study area is generally more uniform, among which the settlements along the rivers and both sides of the roads are larger and denser. The Figure 4 shows the spatial analysis tool of the ArcGIS platform was used to analyze the density of rural settlements in Hongshanyao town, and it can be seen from the figure that there are three high-density areas in Hongshanyao town, namely, Heyanzi village, Yaojiazhai village, and Jiehe village.

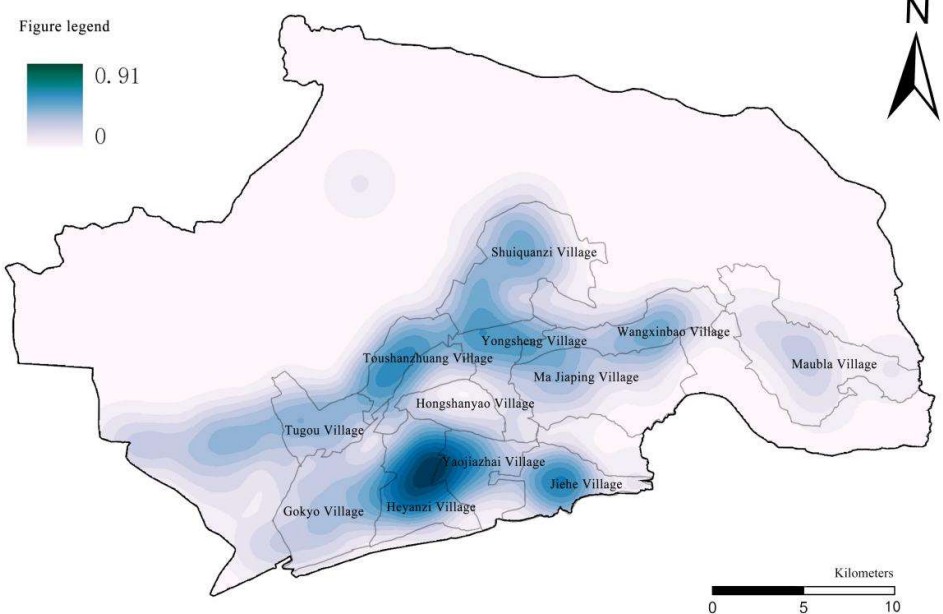

**Figure 4.** Spatial distribution and density of rural settlements in Hongshanyao town.

　　　　The DEM remote sensing image map of Hongshanyao town is processed with the help of the ArcGIS platform, and the analysis of reclassification and overlay shows that most of the rural settlements are distributed in the range of 2000–2500 m above sea level, with a slope of about 2° and flat terrain; the buffer zone is established by combining the rivers, roads and established towns in the study area, and it can be seen from the folding map of rivers, roads, and established towns that the number of settlements is decreasing. The Figure 5 shows the buffer zone map shows that rural settlements in Hongshanyao Town are mostly located within 0–4 km from rivers, and the number of settlements decreases the farther away from rivers; established towns have a positive effect on the development of villages, but most rural settlements are located 4 km away from established towns due to the topography; as for the road buffer zone, rural settlements are mostly located about 2 km away from roads, and the farther away from roads, the more the number of settlements decreases. The farther away from the road, the fewer rural settlements there are.

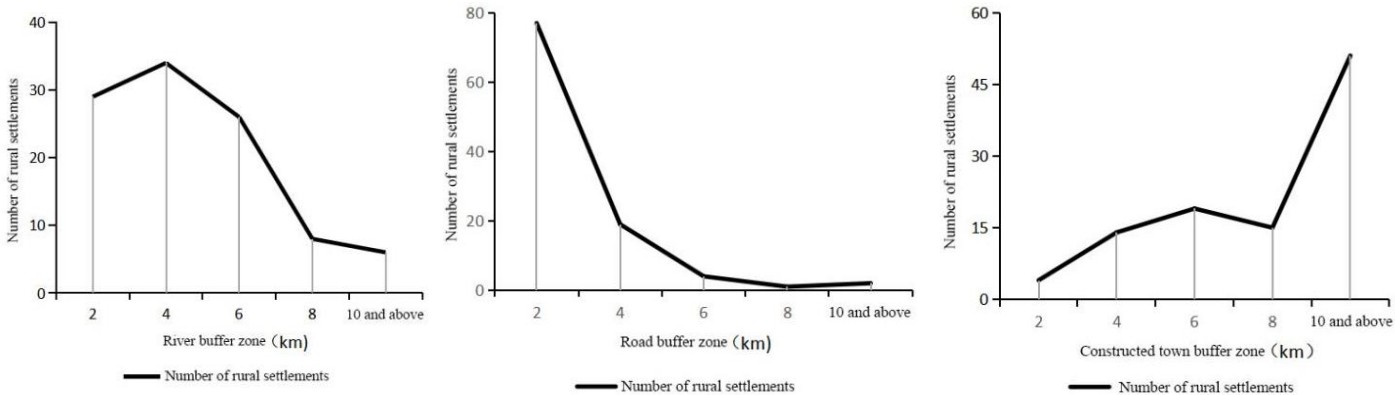

**Figure 5.** Spatial distribution of different buffer zones in Hongshanyao town.

*4.2. Rural Settlement Reconfiguration Model*

　　　　Rural settlement reconfiguration is the rearrangement and optimization of rural spatial patterns under the combined influence of the rural internal development base, external development potential, and villagers' subjective wishes [36,37]. The Figure 6 shows through the factor identification and suitability evaluation of rural settlements, as well as the current situation, characteristics, and development potential of villages in Hongshanyao Township, and combined with farmers' willingness to integrate and relocate, 23 unused houses will be relocated and integrated, and villagers will be guided to move to the advantageous areas for production and living in a reasonable and orderly manner, while the abandoned settlements will be returned to forests for reclamation, to improve the intensification of land use and rural land use. Based on the above analysis of the 12 villages in Hongshanyao Town, they are divided into three categories: gathering and upgrading, suburban integration, and characteristic protection [37].

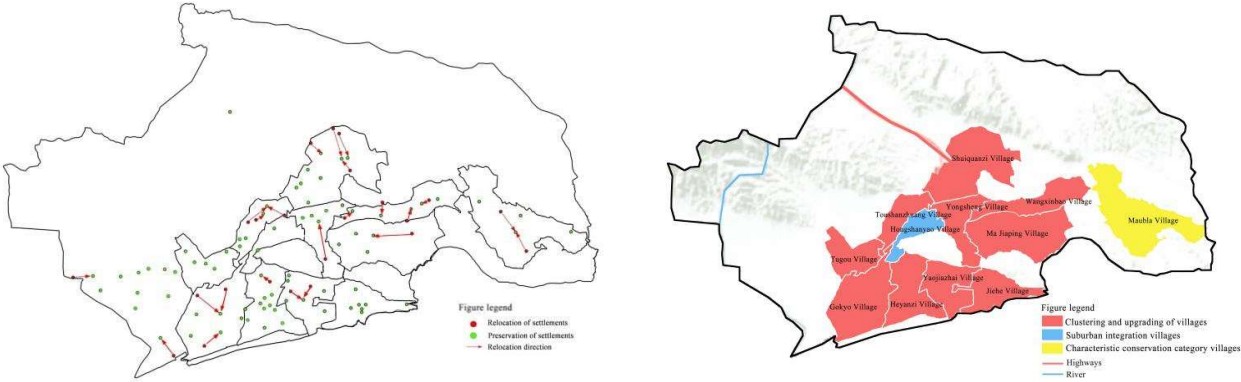

**Figure 6.** Integration of rural settlements in Hongshanyao Township and the reconfiguration model.

(1) Aggregation and upgrading of class

These villages are mainly located in Shuiquanzi Village, Yongsheng Village, Wangxinbao Village, Majiaping Village, Yaojiazhai Village, Heyanzi Village, Gaocheng Village, Tugou Village, Toushanzhuang Village, and Jiehe Village. The development of these villages is mainly based on their characteristic resources and industrial base to promote industrial scale, modernization, and services, promote industrial revitalization, drive economic development, and increase employment opportunities to drive the employment level of the surrounding villages. At the same time, increasing employment can drive the employment level of the surrounding villages, thus promoting the development of villages in a row [38].

(2) Suburban integration class

Villages in the suburban integration category rely on their inherent location advantages, high economic development potential, and convenient transportation, and can provide more convenient services for villagers' lives, mainly in Hongshan Yao village, which is more closely connected with the town and is strongly radiated by it, with relatively complete infrastructure and public service facilities, and is an advantageous area for development within the town, but its development needs to follow the speed of the town's development and share with it, to better promote urban-rural integration. Better promote the integration of urban and rural areas and realize the integrated development of urban and rural areas.

(3) Characteristic protection class

The village of Maubla in the study area is a characteristic conservation village, which is surrounded by mountains and rich in history and culture, and has rich folk culture and historical resources such as the Han and Ming Great Wall sites, and tourism resources such as the national intangible cultural heritage "Swastika lantern". On the one hand, we clarify the value of historical and cultural resources and strengthen their protection; on the other hand, we develop rural tourism to drive the villages with lower economic development, but the development needs to be proportional to prevent the destruction of historical and cultural resources by an excessive developer, resulting in the loss of historical and cultural values. The loss of historical and cultural values [39].

## 5. Conclusions

The text takes the town of Hongshanyao in the northwest arid zone as the research object, based on the full research of the village, based on the subjective willingness of rural villages and the spatial layout of rural settlements, taking the villagers' suitability for living as the entry point, using the method of "SEM model construction + land suitability evaluation", with the logical line of "driving factors -Based on the villagers' questionnaires and the current situation of the villages, we use the model to determine the degree of influence of the farmers on the drivers of rural residential suitability, construct an evaluation system, assign values to different indicators, and analyze the strengths and weaknesses of each village by following per under the requirements of rural revitalization. The evaluation system is constructed to assign values to different indicators, analyze the current situation and strengths and weaknesses of each village, and identify the reconfiguration types and development directions of rural settlements by following the requirements of rural revitalization.

Hongshanyao town, an established town in Jinchang City, has 12 villages with different characteristics. Hongshanyao town has diversified development bases, advantages, and potentials, and the construction and reconstruction models can be selected for different characteristics of villages, which is typical, representative, and can be learned from. As far as Hongshan Yao town itself is concerned, the characteristics are outstanding, but the problems presented carry a certain degree of applicability, and the experience of solving the problems is worth promoting, which is of great significance to the realization of rural revitalization and village planning and construction, and can provide certain help and reference for similar problems in the actual research, which has practical value.

Village development is a long and systematic process, and the village layout should be optimized and progressed with the development of society based on the trend of development and respect for the objective rules of the times. This paper classifies 12 villages in Hongshanyao town of Yongchang County into types, including 10 in the category of gathering and upgrading, 1 in the category of suburban integration, and 1 in the category of characteristic protection. However, there are still some shortcomings in this study, and in future research, village planning should propose guiding strategies in terms of town and village system construction, public services, land remediation, and ecological restoration, to guide the layout and integrated development of villages.

**Author Contributions:** Writing and editing, L.Y.; conceptualization, X.X and L.Y.; methodology, L.Y. and X.X.; funding acquisition, X.X. All authors have read and agreed to the published version of the manuscript.

**Funding:** This work was supported by the National Social Science Foundation of China, grant number 51968037, and the Gansu Provincial Science and Technology Plan Project, grant number 21CX6ZA072.

**Institutional Review Board Statement:** Not applicable.

**Informed Consent Statement:** Informed consent was obtained from all participants involved in this study.

**Data Availability Statement:** Not applicable.

**Acknowledgments:** The authors wish to thank in advance the editor and reviewers for their contribution in the submission and revision phases, most importantly, my supervisor for her guidance.

**Conflicts of Interest:** The authors declare no conflict of interest.

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
