# Peer review of "Reconstructing Rural Settlements Based on Structural Equation Modeling—Taking Hongshanyao Town of Jinchang City as an Example"

_sustainability, doi:10.3390/su15021338_

Round 1

Reviewer 1 Report

1. The research review is too far fetched to comprehensively and clearly sort out the relevant literature supporting this study, and does not reflect the innovation of this study.

2. The applicability analysis of SEM model in residential area research needs to be added.

3. How to reflect the representativeness and universality of this research case? It is suggested to strengthen discussion.

Author Response

Dear Editors and Reviewer,

Thank you for your comments concerning our manuscript entitled “Exploration of Rural Settlement Reconfiguration Model Based on SEM Model and Suitability Evaluation*--Taking Hongshanyao Town of Jinchang City as an example” (sustainability-2113852). These comments are all valuable and very helpful for revising and improving our paper. We have studied comments carefully and have made corrections which we hope meet with approval. Revised portions are marked in red on the paper. The primary corrections in the paper and the responses to the reviewer’s comments are as follows:

Response to Reviewer 1 Comments:

  1. "The research review is too far fetched to comprehensively and clearly sort out the relevant literature supporting this study, and does not reflect the innovation of this study."

Response:Thank you very much for your comments and recognition. In the revision of the article, we have made some changes according to your comments. We have revised the research synthesis, firstly, analyzed the trends and problems of international and domestic rural development, which are currently studied by experts and scholars in various different fields through different perspectives, secondly, summarized the research on rural reconstruction through different ideas, and finally, presented the purpose, significance and value of the article research through the perspective of rural revitalization.

  1. "The applicability analysis of SEM model in residential area research needs to be added."

Response:Thank you very much for your comments and recognition. In the revision of the article, we have made some changes based on your comments. We introduced the addition of the interpretation of the applicability of the model to rural habitats at the beginning of the research methodology, the SEM model to enhance the stability of the results and the multi-factor analysis of rural settlements.

  1. "How to reflect the representativeness and universality of this research case? It is suggested to strengthen discussion."

Response:Thank you very much for your comments and recognition. In the revision of the article, we have made some changes based on your comments. This study case is selected from Hongshan Yao town, which consists of 12 administrative villages, each of which has different characteristics, development potentials and development types. There are different topographies and different types of industries in the study area, with diverse development bases, advantages and potentials, and the study area case is special and typical, also in the northwest of China, the characteristics of each village in Hongshan Yao town almost encompass the villages in this area Therefore, this case has certain representativeness and universality.

Special thanks to you for your good comments.

We tried our best to improve the manuscript and made some changes in the manuscript.  These changes will not influence the content and framework of the paper. And here we did not list the changes but marked them in red in the revised paper.  

We appreciate your warm work earnestly and hope that the correction will meet with approval.  

Once again, thank you very much for your comments and suggestions.

Yours sincerely,

Lin Ye

Reviewer 2 Report

I consider this paper an important contribution to current literature. The method validation has the explanation for understanding the aim exposed.

I do not have any comment on this version.

Author Response

Dear Editors and Reviewer,

Thank you for your comments concerning our manuscript entitled “Exploration of Rural Settlement Reconfiguration Model Based on SEM Model and Suitability Evaluation*--Taking Hongshanyao Town of Jinchang City as an example” (sustainability-2113852). These comments are all valuable and very helpful for revising and improving our paper. We have studied comments carefully and have made corrections which we hope meet with approval. Revised portions are marked in red on the paper. The primary corrections in the paper and the responses to the reviewer’s comments are as follows:

Response to Reviewer 2 Comments:

"I consider this paper an important contribution to current literature. The method validation has the explanation for understanding the aim exposed.

I do not have any comment on this version."

Response:Thank you very much for your comments and recognition. In order to make the article more research-oriented, we have revised the abstract, review, and explanatory notes on the applicability of the model, and strengthened the representativeness and generality of the cases in the discussion section of the article.

Special thanks to you for your good comments.

We tried our best to improve the manuscript and made some changes in the manuscript.  These changes will not influence the content and framework of the paper. And here we did not list the changes but marked them in red in the revised paper.  

We appreciate your warm work earnestly and hope that the correction will meet with approval.  

Once again, thank you very much for your comments and suggestions.

Yours sincerely,

Lin Ye

Reviewer 3 Report

General Comments:

This is an interesting work on the “investigation of Hongshanyao rural settlement reconfiguration using Structural Equation Model and the suitability of the models developed”. This article is quite concise which is good. However, the title of this article is quite vague, and the abstract is difficult to understand due to the unclear flow of explanations. The methods used in this study are suitable, although clarifications of some details are required. Below are some specific comments.

Specific Comments:

Title: The title “Exploration of Rural Settlement Reconfiguration Model Based on SEM Model and Suitability Evaluation*--Taking Hongshanyao Town of Jinchang city as an example” is not a good title for an article. This title is more like an explanation which should be in the materials and method section. I suggest you change the title to “Investigation of Hongshanyao Rural Settlement Reconfiguration Using Structural Equation Model and The Suitability of The Models Developed”. OR you can change it to another title that can make it easy for your readers to understand.

Abstract: I suggest you re-write the whole of the abstract section due to the unclear flow of explanations. Also, in this type of abstract, it would not be a good idea to number the points in your abstract. Instead of numbering (1, 2, 3) your points, you can use the “comma” and “and”.

For example; you can change “The conclusions show that (1) among the 529 farm households included, it is found through the research that most of them have the willingness to spatially reconstruct the settlement; (2) the conditions of farmers' disposable income, living environment, and infrastructure have a strong influence on the satisfaction of farm households' residence; (3) the reconstruction and optimization of rural settlements are based on the intrinsic development foundation of rural areas, external development potential, and villagers' subjective willingness.” to

“The conclusions show that among the 529 farm households included, it is found through the research that most of them have the willingness to spatially reconstruct the settlement, the conditions of farmers' disposable income, living environment, and infrastructure have a strong influence on the satisfaction of farm households' residence, and the reconstruction and optimization of rural settlements are based on the intrinsic development foundation of rural areas, external development potential, and villagers' subjective willingness.” However, I would suggest you should paraphrase the whole sentence.

P1, Line 13, change “wishes” to “needs”

Introduction: P1 and 2, Line 42 – 66, The in-text referencing was done wrongly. Please correct your in-text referencing style.

P2, Line 90 – 91, Correct the repetitions there.

P3, Line 97, ..to explore the development model with different characteristics.8. Remove 8.

P3, Line 99, change “…576 of which were returned, of which 529 were valid, with a sample efficiency of 88%.” to “…576 were returned, of which 529 were valid, with a sample efficiency of 88%.”

P3, Line 105 – 106, The Table 1 is quite confusing because of the addition of Jinchang and Yongchang. I would suggest you remove these city and county names from your table and include only the township (Hongshanyao). This is because Hongshanyao is your main study focus and it makes it easier to understand. It is good you mentioned it in your texts, but I do not suggest you include it in your table.

P12, Line 278 – 282, “Based on the above analysis of the 12 villages in Hongshanyao Town, they are divided into three categories: gathering and upgrading, suburban integration and characteristic protection [38,39]. The above analysis of the 12 villages in Hongshanyao Township will be divided into three categories: gathering and upgrading, suburban integration, and characteristic protection.” I am not sure of what you mean. Is this a repetition? Please paraphrase your sentence for easy comprehension.

Overall, this is a good research study. However, I suggest you do a major edit to this article. 

Author Response

Dear Editors and Reviewer,

Thank you for your comments concerning our manuscript entitled “Exploration of Rural Settlement Reconfiguration Model Based on SEM Model and Suitability Evaluation*--Taking Hongshanyao Town of Jinchang City as an example” (sustainability-2113852). These comments are all valuable and very helpful for revising and improving our paper. We have studied comments carefully and have made corrections which we hope meet with approval. Revised portions are marked in red on the paper. The primary corrections in the paper and the responses to the reviewer’s comments are as follows:

Response to Reviewer 3 Comments:

Comment: Title: The title “Exploration of Rural Settlement Reconfiguration Model Based on SEM Model and Suitability Evaluation*--Taking Hongshanyao Town of Jinchang city as an example” is not a good title for an article. This title is more like an explanation which should be in the materials and method section. I suggest you change the title to “Investigation of Hongshanyao Rural Settlement Reconfiguration Using Structural Equation Model and The Suitability of The Models Developed”. OR you can change it to another title that can make it easy for your readers to understand.

Response:We are sorry for the inappropriate expressions and language of the topic that is difficult to understand. After your comments, we have sorted out and thought about the main content of the article, and now we have revised the title to: Reconstructing Rural Settlements Based on Structural Equation Modeling - Taking Hongshan Yao in Jinchang City as an Example, which is more brief than the previous title and more able to highlight the focus of the article.

Comment:Abstract: I suggest you re-write the whole of the abstract section due to the unclear flow of explanations. Also, in this type of abstract, it would not be a good idea to number the points in your abstract. Instead of numbering (1, 2, 3) your points, you can use the “comma” and “and”.

Response: We are very sorry for our inappropriate expression. This article focuses on the different types of development in rural areas that can help promote rural revitalization under the impact of urbanization, contributing methodologically and analytically, and trying to inform spatial optimization from a rural perspective. In the revision, we have rewritten the abstract to make it easier to read, as follows.

The rapid urbanization has driven great changes in China's rural areas. In order to adapt to the changes in the internal elements and external regulation of the countryside, rural reconfiguration, i.e., to make rural development more adaptable to the spatial pattern of social development through optimal allocation and effective management, can achieve structural optimization and functional improvement within the rural territorial system. This study selects Hongshanyao Town of Jinchang City as the study area, constructs a structural equation model and an adaptability evaluation system to investigate the suitability of rural settlement layout, and constructs a differentiated and suitable rural settlement reconstruction model for different villages based on meeting farmers' wishes, to intending to serve the current major national strategic needs and solve the real dilemmas faced by rural areas in the process of urban-rural transformation and development. The corresponding strategies are proposed, which provide a theoretical basis for future village development and spatial reconfiguration practices in rural areas, and are of great significance for realizing rural revitalization as well as village planning and construction. (Lines 8-20)

Comment: P1, Line 13, change “wishes” to “needs”.

Response: Thank you very much for your comments. In the revision of the article, we have rewritten the abstract in accordance with the previous comment.

Comment: Introduction: P1 and 2, Line 42 – 66, The in-text referencing was done wrongly. Please correct your in-text referencing style.

Response: Thank you very much for your suggestion, and I apologize for the oversight in the citation of the article. Currently, in the revision of the article, we have reorganized the literature review to better present the significance and background of the study, and also corrected the citation errors.

Comment: P2, Line 90 – 91, Correct the repetitions there.

Response: Thank you very much for your suggestion and we are very sorry for our oversight. In the revision, we have removed this duplicate paragraph.

Comment: 3, Line 97, ..to explore the development model with different characteristics.8. Remove 8.

Response: Thank you very much for your suggestion and we are very sorry for our oversight. In the revision, we have removed this 8.

Comment: P3, Line 99, change “…576 of which were returned, of which 529 were valid, with a sample efficiency of 88%.” to “…576 were returned, of which 529 were valid, with a sample efficiency of 88%.”

Response: Thank you very much for your suggestions, we have made changes according to your suggestions as follows.

In this survey, a total of 12 administrative villages in one town were selected as sample points. 600 questionnaires were distributed, and 576 were returned, of which 529 were valid, with a sampling efficiency of 88%.

Comment: P3, Line 105 – 106, The Table 1 is quite confusing because of the addition of Jinchang and Yongchang. I would suggest you remove these city and county names from your table and include only the township (Hongshanyao). This is because Hongshanyao is your main study focus and it makes it easier to understand. It is good you mentioned it in your texts, but I do not suggest you include it in your table.

Response: Thank you very much for your suggestions. We have modified the table according to your suggestions and deleted the names of the cities and counties in the table for clarity, as follows.

Table 1. Distribution of the survey sample.

Village

Number of samples

Effective sample size

Hongshanyao Village

50

48

Shantouzhuang Village

50

45

Shuiquanzi Village

50

42

Yongsheng Village

50

46

Wangxinbao Village

50

41

Ma Jiaping Village

50

42

Jiehe Village

50

48

Yaojiazhai Village

50

43

Heyanzi Village

50

44

Gokyo Village

50

44

Tugou Village

50

45

Maubla Village

50

41

Total

600

529

Comment: P12, Line 278 – 282, “Based on the above analysis of the 12 villages in Hongshanyao Town, they are divided into three categories: gathering and upgrading, suburban integration and characteristic protection [38,39]. The above analysis of the 12 villages in Hongshanyao Township will be divided into three categories: gathering and upgrading, suburban integration, and characteristic protection.” I am not sure of what you mean. Is this a repetition? Please paraphrase your sentence for easy comprehension.

Response: Thank you very much for your suggestion and we are very sorry for our oversight. In the revision, we have removed this duplicate paragraph.

We tried our best to improve the manuscript and made some changes in the manuscript.  These changes will not influence the content and framework of the paper. And here we did not list the changes but marked them in red in the revised paper.  

We appreciate your warm work earnestly and hope that the correction will meet with approval.  

Once again, thank you very much for your comments and suggestions.

Yours sincerely,

Lin Ye

Round 2

Reviewer 1 Report

The author has revised the comments one by one and agreed to publish them

Reviewer 3 Report

Dear Authors,

You have done a good job.